# Deep learning-based classification of peptide analytes from single-channel nanopore translocation events

**Bryan A. Krantz** *

Department of Microbial Pathogenesis, School of Dentistry, University of Maryland, Baltimore, Maryland, United States of America

* bkrantz@umaryland.edu

## Abstract

Rapid and accurate detection of peptide biomarkers using nanopore biosensors is critical for disease diagnosis and other biomedical applications. Processing large, complex single-channel translocation data streams poses a significant challenge for peptide analyte classification. Here, we present a supervised deep learning data processing pipeline for peptide classification from translocation events. The first stage employs a convolutional and recurrent neural network, adapted from the Deep-Channel multi-channel classifier, to accurately classify raw current recordings into discrete conductance states, including partially blocked sub-conductance intermediates. The second stage, peptide classification, utilizes a novel branched input network with a temporal convolutional network for processing translocation event conductance state sequences and a dense network for incorporating computed event-level and global kinetic features. Using idealized simulated multi-state translocation data for seven peptides, we demonstrate high classification accuracy (0.9998 (±0.0006)) when global features are included alongside event-level features. For classifying mixture samples, where only event-level features are applicable, performance shows more modest accuracy (0.70 (±0.01)). Peptide mixture predictions showed reasonable accuracy (MAE 0.045–0.161), although misclassification resulted in false positives. Event stochasticity and the fact that some peptides possessed similar kinetic parameters posed challenging for event-level prediction. However, vote aggregation from translocation event streams achieves perfect 100% accuracy, when predicting pure peptide samples. This proof-of-concept study demonstrates a robust deep learning framework for nanopore peptide classification using simulated data, laying the groundwork for classifying peptides from complex mixtures using real experimental data with the anthrax toxin protective antigen nanopore.

**Data availability statement:** All experimental electrophysiological records, peptide translocation event stream simulations, and related source code are publicly available. The datasets used in this manuscript have been deposited in the Zenodo repository under the DOI: 10.5281/zenodo.16965049. The source code is maintained on a GitHub repository (https://github.com/bakrantz/Pept-Class).

**Funding:** National Institutes of Health (1R01AI077703 and 5R21AI124020). The funders had no role in study design, data collection and analysis, decision to publish, or preparation of the manuscript.

**Competing interests:** NO.

## Introduction

Rapid and accurate detection of biomolecules (including nucleic acid polymers, proteins, peptides, and other small molecules) is a critical challenge in timely disease diagnosis. Peptide biomarkers produced during various disease processes (e.g., heart disease, infectious disease, and cancer) may be critical molecular targets to be sensed to aid in proper diagnosis [1–3]. Nanopore peptide biosensors combined with powerful single-molecule detection offer a potential solution to this problem [4]. Moreover, this technology also may lead to innovations in related biomedical areas, including drug discovery [5], biopolymer sequencing [6–8], and basic science research.

Nanopore biosensors (reviewed [8]) are comprised of two aqueous compartments bathed in electrolyte solutions, which are separated by a thin membrane. In the case of biological protein nanopores, the membrane is a planar lipid bilayer, where the bilayer-inserted protein nanopore creates a nanometer-scale pore (or channel), which spans the membrane. In the case of solid-state nanopores, these systems utilize a membrane generally made of a substrate like silicon nitride [7], glass [9], or graphene [10], where a small nanometer-scale aperture is created in the membrane. Under an applied voltage, biomolecules can translocate through these pores, and their passage causes a measurable change in current as they impede the flow of ions. These small picoamp-scale changes in ionic current are easily detected by voltage-clamp amplifiers even in the single-molecule limit. When a single pore is present in the system, a single-channel recording of a stream of biomolecule analyte translocations is obtained.

Many protein nanopores have been investigated as potential biosensors, including α-hemolysin [11–14], aerolysin [12,15,16], *Mycobacterium smegmatis* porin A [17], and the curli production assembly/transport component (CsgG) [18,19]. Like the dedicated protein translocase, CsgG, anthrax toxin (Fig 1A) is a protein translocase model system [20] with potential to be a peptide biosensor platform. It is a tripartite protein toxin made up of a homooligomeric nanopore channel, called protective antigen (PA), which translocates the two other component enzymes, lethal factor (LF) and edema factor. Being a robust protein translocase, anthrax toxin can be adapted to translocate heterologous proteins [21] and peptides [22,23] with no required substrate modifications, like DNA tags. Larger peptides [24,25] and proteins [26] translocate via a muti-step process through a series of fully blocked intermediates, whereas shorter peptides translocate through multiple conductance state intermediates [22] (Fig 1B), suggesting potential peptide discriminating features that may be exploited in biosensing applications.

Many large, nuanced single-channel datasets may be generated when implementing a nanopore biosensor system. Inevitably there will be long seconds-to-minutes recordings of many stochastic translocation events; and often numerous recordings are needed to encompass potential analytes of interest. The translocation recordings themselves may not be simply two conductance states (bound to pore/fully blocked and open pore), but rather there may be many conductance states, including partially-blocked sub-conductance species (Fig 1B) [22]. To enhance the ability to

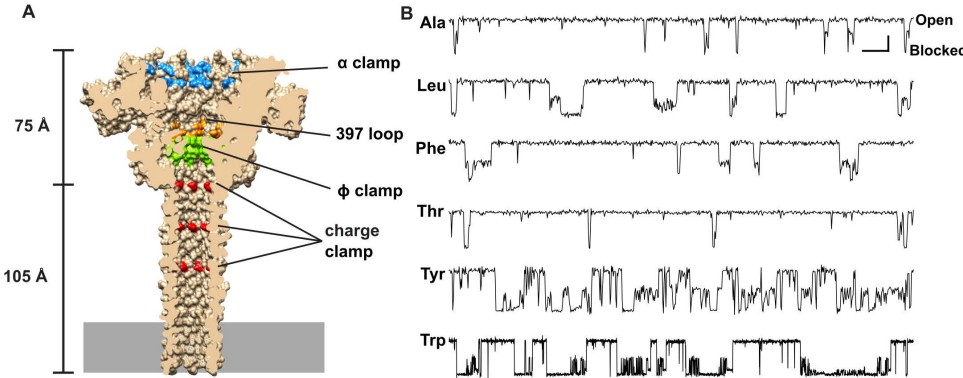

**Fig 1. Anthrax toxin PA as potential nanopore peptide biosensor platform. (A)** Cryo-electron microscopy structure of the PA nanopore [27] showing the locations of key peptide interacting active sites, clamps, and loops (labeled and color coded). Dimensions of its globular cap and elongated β-barrel stem are indicated. Membrane bilayer is depicted by gray rectangle at bottom of β barrel. **(B)** Example single-channel peptide translocation events via the PA nanopore at 70 mV driving force. Analytes are a 10-residue guest-host peptide (KKKKKXXSXX), where the guest residues (X) are indicated at left of each record. The system populates multiple discrete conductance state intermediates during translocation events; current levels of fully open and fully blocked nanopores are labeled at top right. Scalebar given at upper right is 2 pA by 100 ms for X = A, L, T, F, Y and 2 pA by 500 ms for X = W.

distinguish analytes, multiplex approaches many be taken, where multiple different pores (or even engineered ones) are used to gain additional read-outs of a particular analyte.

To process these types of large complex data streams from ion channels or nanopores and correctly classify analytes, machine learning or deep learning computational approaches can be taken [16,28–31]. Deep learning methods utilize neural networks (NN) of various architectures and layerings, which may be trained to find patterns in input single-channel recordings of translocation event streams, leading ultimately to output classifications. These NN approaches are highly modular and adaptable, where the temporal patterns in the translocation event state sequence and their corresponding computed features can be successfully learned. Here we describe a generalizable deep learning approach to first classify conductance states of a multi-state protein translocase nanopore (anthrax toxin PA) and second classify simulated peptides based on their translocation events, as they populate conductance state intermediates.

## Materials and methods

### Proteins

Heptameric PA oligomer (PA$_7$) was prepared as described [32]. Briefly, full-length PA$_{83}$ (83 kDa) was expressed in *Escherichia coli* BL21(DE3) using a pET22b plasmid directing expression to the periplasm. PA$_{83}$ was extracted from the periplasm and purified using Q-Sepharose anion-exchange chromatography in 20 mM Tris-Cl, pH 8.0, and eluted with a gradient of 20 mM Tris-Cl, pH 8.0 with 1 M NaCl. PA$_{83}$ was then treated with trypsin (1:1000 wt/wt trypsin:PA) for 30 min at room temperature to form nicked PA. The trypsin was inhibited with soybean trypsin inhibitor at 1:100 dilution (wt/wt soybean trypsin inhibitor:PA). The trypsin-nicked PA was subjected to Q-Sepharose chromatography to isolate the oligomerized PA$_7$ by applying the trypsin-nicked PA to the Q-Sepharose column in 20 mM Tris-chloride, pH 8.0. The oligomerized PA$_7$ was eluted by a gradient of 20 mM Tris-Cl, 1 M NaCl, pH 8.0. A chimeric construct of the 30-residue amino-terminal leader sequence of LF and Colicin-E7 immunity protein from *E. coli* (IM7) was created using PCR as described [33]. The construct was purified by His6-affinity chromatography identically to the procedure used for LF's amino-terminal domain

as described [34]. Ten-residue guest–host peptides were synthesized by Elim Biopharmaceuticals without further purification as described [22,23].

## Single-channel electrophysiology

Planar lipid bilayer currents were recorded with an Axopatch 200B amplifier and a Digidata 1440A acquisition system (Molecular Devices Corp.) as described [22,32,34]. Membranes were painted on a 50-µm aperture of a 1-mL white Delrin cup with 3% (wt/vol) 1,2-diphytanoyl-$sn$-glycero-3-phosphocholine (Avanti Polar Lipids) in $n$-decane. The $cis$ (side to which the $PA_7$ is added) and $trans$ chambers were bathed in universal bilayer buffer (10 mM oxalate, 10 mM phosphate, 10 mM Mes, 1 mM EDTA, 100 mM KCl, pH 5.6). Single-channel recordings were filtered at 400 Hz using a multi-section Bessel filter and recorded at 800 Hz using PCLAMP10 software. The applied voltage is defined as $\Delta\psi = \psi_{cis} - \psi_{trans}$ (with $\psi_{trans}$ set to 0 mV).

Single-channel recordings of a guest-host peptide and IM7 translocations were carried out as described [22] and used as training data for three-state and two-state conductance state classification, respectively. A single PA channel was inserted into a painted bilayer at a $\Delta\psi$ of 30 mV by adding ~2 pM of $PA_7$ (freshly diluted from a 2-µM stock) to the $cis$ side of the membrane. Once a single channel inserted, the $cis$ chamber was perfused by fresh buffer to remove excess uninserted $PA_7$. Then the desired peptide/protein analyte to be translocated was added to the $cis$ chamber at 20–100 nM. Translocation data were acquired by stepping the applied $\Delta\psi$ to a higher positive value and collecting recordings of the translocation event stream for several minutes. Translocation recordings were subsequently labeled for two or three discrete conductance states in CLAMPFIT (ground truth used in NN training), where short-duration spikes were ignored.

## Hardware, software, and environment used for deep learning

We used Anaconda to create a Python 3.9 environment, where TensorFlow [35] (2.12.0), Keras Temporal Convolutional Network (TCN) [36] (3.5.6), and other standard modules were installed. At first, a 2014 MacBook Air was used in NN training and prediction. To improve training performance, this hardware was upgraded to a 2025 MacBook Pro with M4 Apple Silicon and 24 GB of RAM. GPU cores were used during training on that device by installing tensorflow-metal. All software source code is available on GitHub (https://github.com/bakrantz/Pept-Class).

## Simulated peptide translocation records

Peptide translocations through protein nanopores were simulated assuming a stochastic Markovian process, where transition from one species to another occurred by sampling a probability transition matrix. Simulations were sampled at 1000 Hz for 30 s for peptides A-F and 150 s for peptide G, allowing similar numbers of translocation events to occur. There were three conductance states in all simulated peptides, i.e., fully blocked by peptide (state 0), partially blocked by peptide (state 1), and fully conducting (state 2). Peptides A-F possessed exactly three species corresponding to each of those conductance states. Peptide G, on the other hand, had two different fully blocked state 0 conductance states alongside species directly corresponding to state 1 and state 2, and therefore, had four total species. The conductance states versus time simulations were ideal in these simulations with no added noise. The simulations revealed exponential cumulative distribution functions (CDF) of dwell times for state 0 or state 1 for the segmented translocation events, albeit Peptide G's state 0 dwell times were best fit to a double exponential decay reflecting the presence of two state 0 species. Simulated data was chosen in this study to have a large enough controlled dataset to train robust deep learning models and facilitate their development.

## Classification of multi-conductance-state channels

The Convolutional Neural Network-Recurrent Neural Network (CNN-RNN) model implemented in Deep-Channel [28], which classifies multi-channel patch-clamp current versus time records, was used in the multi-state (and binary) conductance state classifiers presented here. Briefly, a deep neural network was built with TensorFlow and Keras [35], where the model architecture consisted of a hybrid CNN and Long Short Term Memory (LSTM) network designed to learn temporal

dependencies within the single-dimensional current versus time data. Specifically, the scaled current input data, with a single feature, was first processed by a time distributed layer wrapping a 1D convolutional layer. This layer employed 64 filters with a kernel size of one and rectified linear unit (ReLU) activation to extract local features at each time step. Following the convolutional layer, a time distributed MaxPooling1D layer with a pool size of one was used for downsampling. The output was then flattened to prepare the features for the recurrent layers.

The flattened temporal features were then fed into a stack of three LSTM layers, each with 256 units and ReLU activation. The first two LSTM layers were configured to return sequence, allowing them to feed into the subsequent LSTM layer. Each LSTM layer was followed by a batch normalization layer for stabilizing training and a dropout layer with a rate of 0.25 to prevent overfitting. The final LSTM layer outputted a single sequence which was then passed through a batch normalization and a Dropout layer. The output consisted of a Dense layer of units equal to the number of conductance states using a softmax activation function. This provided a probability distribution over the possible conductance states for each input current value.

The model was trained using the categorical cross-entropy loss function, suitable for multi-class classification with categorical labels. The Adam optimizer was used to update the network weights during training. Performance was evaluated using accuracy, precision, recall, and macro-averaged F1-score. The model was trained for 15 epochs with a batch size of 32, and validation data was used to monitor performance during training. At the completion of training, model weights were saved for future predictions.

The main differences between the single-channel conductance state classifier implemented here and Deep-Channel is the dropout in each LSTM layer was increased to 0.25 and the Adam optimizer was used (instead of stochastic gradient descent) to update weights during training. Also, of course, Deep-Channel classifies number of channels, whereas our classifier trains for the discrete conductance states populated by a single nanopore.

### Deep learning-based peptide classifier

The classification of peptides based on their single-channel translocation events was performed using a branched NN architecture implemented in TensorFlow and Keras [35]. This model was designed to process two distinct types of input derived from each translocation event: (i) the sequence of discrete conductance states observed during the peptide's passage through the nanopore, represented in our test example as a series of state 0 or state 1 time points; and (ii) a set of computed kinetic features characterizing the translocation event conductance state sequences.

Two different groups of features were differentially utilized depending on the peptide classification application: (i) 13 event-level features; and (ii) 7 global features computed over an entire pure peptide translocation event stream. The initial set of event-level features included: state sequence entropy, first transition time, average dwell in state 0, average dwell in state 1, variance of dwell in state 0, variance of dwell in state 1, longest dwell in state 0, longest dwell in state 1, event duration, probability in state 0, probability in state 1, ratio of probabilities in state 0 to state 1, and number of transitions. Note we removed entropy from the initial set when permutation importance analysis showed that its inclusion did not improve F1-scores. The initial set of global features included: average of event duration, variance of event duration, average event entropy, average first transition time, average number of transitions, overall probability in state 0, overall probability in state 1, and overall ratio of probability in state 0 to state 1. The global feature, variance of event duration, was removed when it significantly degraded performance, as described in the Results. When event-level training was employed to ultimately predict mixtures of peptides in an event stream recording, then only the event-level features are used. But when a pure peptide stream requires classification, then both global and event-level features can be used. The basic network architecture is similar in either instance, as described below.

The simulated records of peptide A-G translocation event streams were segmented in a separate routine into a list of translocation state sequences alongside their corresponding features. Very short events (< 5 ms) were filtered out during segmentation by a user-defined parameter so the model could train on more information rich longer events. These

segmented translocation events were organized as a Python list of dictionaries and saved as pickle files for proper loading in the training and prediction scripts. At the loading of the pickle files during training, an *in situ* downsampling was performed as needed to maintain class balance.

Immediately prior to training the input translocation event conductance state sequences were padded to a uniform length (determined by the maximum sequence length in the training set) using a value of −1.0. (This padding value was chosen to distinguish it from the state sequences, which were composed of state 0 or state 1 values). The features were standardized using a StandardScaler fitted on the training data. This scaler object was saved for later use in prediction.

During training, the conductance state sequence of each translocation event was fed into a TCN [36] branch. ReLU activation functions were employed in the dense layers of this branch to introduce non-linearity. This branch consisted of two sequential TCN blocks. The first TCN block comprised 256 filters with a kernel size of 3 and dilation rates of 1, 2, 4, followed by batch normalization and a dropout layer with a rate of 0.3. The second TCN block similarly utilized 128 filters, a kernel size of 3, dilation rates of 1, 2, 4, batch normalization, and a dropout rate of 0.3. The output of the second TCN block was then processed by a global average pooling 1D layer to produce a fixed-size vector representation of the conductance state sequence. The number of TCN blocks, dilations, filters and kernel size were chosen to balance the ability to maintain evaluation metrics while reducing computational costs.

The second input branch processed the computed kinetic features, which were derived from the translocation event state sequences. The number and type of features, global and/or event-level) used depended on the ultimate classification application. This branch consisted of a dense layer with 32 units and ReLU activation, followed by batch normalization and a dropout layer with a rate of 0.3. The outputs of the TCN branch and the Dense features branch were then concatenated. This merged representation was fed into a final series of dense layers: a dense layer with 64 units and ReLU activation, followed by a dropout layer with a rate of 0.3, and finally, an output dense layer of units equal to the number of peptide classes, utilizing a softmax activation function to yield a probability distribution over the peptide identities. In general, dropout regularization was employed throughout to reduce overfitting.

The model was compiled using the Adam optimizer with a learning rate of 0.001 and the categorical cross-entropy loss function. Model performance was monitored using accuracy, precision, recall, and macro-averaged F1-score. Training was conducted with a batch size of 32 for 30 epochs, with 20% of the training data reserved for validation. To prevent overfitting and optimize training, early stopping (patience of 20 epochs), model checkpointing (saving the best weights based on validation loss), and a learning rate reduction on plateau (factor of 0.5, patience of 5 epochs, minimum learning rate of $10^{-5}$) were implemented as callbacks during the training process.

The selection of the TCN/Dense architecture's specific parameters, including the number and configuration of TCN blocks, filter counts, dilation rates, and kernel sizes, as well as the dropout rate, was refined through an iterative manual tuning process. This involved systematic experimentation and evaluation on the validation dataset to optimize for classification performance and training stability.

Training and evaluation of the model was performed for multiple replicates at different train test splits of the peptide translocation event input data by setting the random_state parameter to different values. Means and standard deviations of the evaluation metrics were computed from those replicates.

## Mixed peptide sample prediction

The ability of the trained peptide classifier to identify components of a mixed peptide sample was evaluated. A synthetic mixed sample of translocation events was generated by randomly selecting a defined number of events from the individual peptide datasets according to pre-set fractional compositions (e.g., 60/40 Peptide A/Peptide D mixture would sample these respective pure datasets in the proper ratio to make 500 total events). The features scaler and weights saved from translocation event training using only event-level features were loaded into the same branched input TCN/Dense model. Predictions using the model outputted a probability distribution over the seven possible peptide identities for each input

translocation event. To obtain a classification for each event, a confidence threshold of 0.20 was applied to the predicted probabilities. If the maximum predicted probability for an event exceeded this threshold, the event was assigned to the corresponding peptide class. Events with maximum predicted probabilities below the threshold were considered unclassified (called 'None'). This confidence threshold can be adjusted as needed to obtain more confident predictions and can help minimize false positives. Following the event-level classification, the overall composition of the mixed sample was estimated by counting the number of confident predictions for each peptide class. Evaluation metrics such as Mean Absolute Error (MAE), Mean Squared Error (MSE), and Root Mean Squared Error (RMSE) were calculated to quantify the difference between the predicted peptide fractions and actual fractions.

### Vote total prediction of pure peptides

For the evaluation of pure peptide translocation event streams, a "vote total" approach was employed. Here, instead of solely classifying individual translocation events, the entire stream of events corresponding to a single peptide was processed. Each individual translocation event within the stream (consisting of its conductance state sequence and the 12 kinetic features) was passed through the previously trained TCN/Dense NN. The "vote" for each event was determined as the peptide class with the highest predicted probability. Following the prediction for every event in the stream, the total number of votes for each of the seven peptide classes was tallied. The peptide class receiving the highest number of votes was considered the predicted identity of the entire stream of translocation events. Several metrics were calculated to assess the performance of this vote total prediction method for each test peptide: Top-1 Accuracy (binary metric indicating whether the peptide class receiving the most votes matched the known identity of the peptide stream), Confidence (fraction of total votes received by the winning peptide class out of the total number of events in the stream), Vote Entropy (Shannon entropy of the vote distribution using base 2 log), and Rank of True Label (rank of the true peptide label based on the vote counts).

## Results

### Deep learning pipeline

To develop a robust pipeline to analyze single-channel peptide translocation through nanopores, we aimed to develop (i) a single-channel conductance state classifier and then (ii) a peptide analyte classifier both using deep learning techniques. We first created a NN to classify the conductance state of current recordings based on the CNN-LSTM architecture in the multi-channel classifier, Deep-Channel [28]. We re-purposed Deep-Channel's TensorFlow/Keras-based NN implementation in Python 3.9 to at first classify a binary system of open or blocked two-state channels (often occurring when larger proteins and peptides translocate via anthrax toxin). We achieved excellent predictive labeling of conductance state using both simulated two-state data and realistic protein translocation data from anthrax toxin. The binary classification training of the two-state anthrax toxin protein translocation data achieved high performance, with a test set accuracy of 0.995, a precision of 0.998, a recall of 0.994, and a Matthews Correlation Coefficient (MCC) of 0.988, indicating excellent discrimination between the two conductance states. Then we moved to multi-state classification of conductance state (the situation germane to peptide analyte translocation via anthrax toxin PA) using the same Deep-Channel CNN-RNN architecture but with an output appropriate for multiple conductance states. In our benchmark conductance state classifications, we again used a simulated three-state current recording as well as a realistic three-state peptide analyte translocation via anthrax toxin (Fig 2A). The multistate conductance prediction for anthrax toxin peptide translocation data achieved high accuracy, with an overall multi-class accuracy of 0.989 (Fig 2B,C). The macro-averaged precision, recall, and F1-score were also high at 0.968, 0.964, and 0.965, respectively, indicating good performance across all three conductance states (fully open, partially blocked intermediate, and fully blocked). The confusion matrix shows strong classification for the fully open state (class 2), with some misclassification between the fully blocked (class 0) and partially blocked (class 1) states.

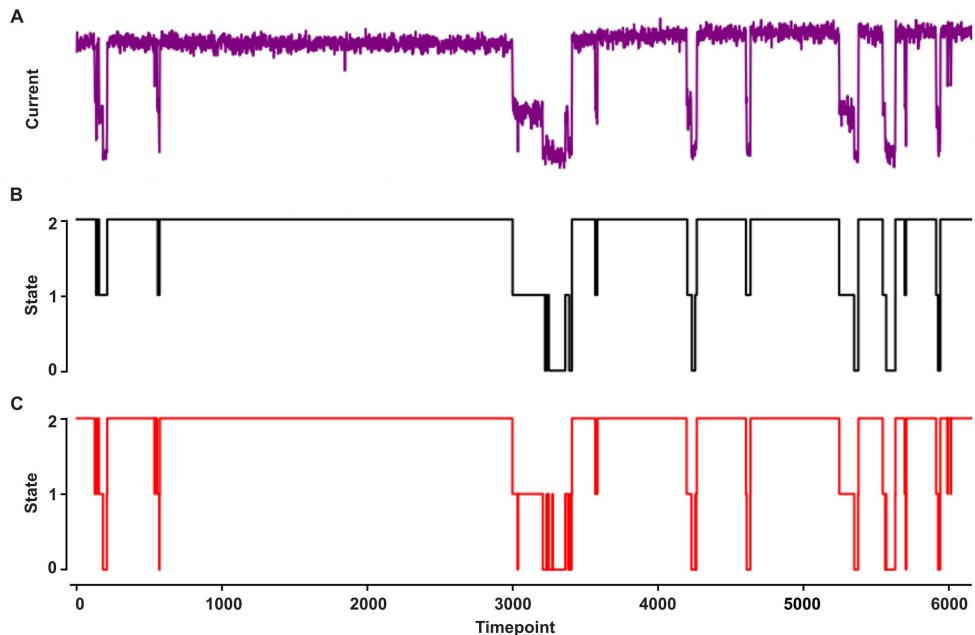

**Fig 2. Conductance state classification.** A three-state guest-host peptide translocating via anthrax toxin PA. **(A)** Raw single-channel current record (purple trace), where open nanopore current is ~3 pA and time is sampled at 800 Hz. **(B)** Ground truth labeling of conductance state (black trace) using CLAMPFIT: fully peptide blocked nanopore (state 0), partially peptide blocked nanopore (state 1), and fully open nanopore (state 2). Note some short-duration events were ignored. **(C)** Deep learning CNN-RNN model prediction of conductance state (red trace).

It should be noted the CNN-RNN model predicted some fast-flickering events (e.g., state 1→0) in the anthrax toxin data-set that were not classified as such by ClampFit (our ground truth classification) as those very short events were set to be ignored (Fig 2B), showing the deep learning approach includes sensitivity for rapid events.

## Simulated multi-state peptide translocation datasets

To explore the utility of using deep learning to classify peptides from their multi-state translocation events, we simulated seven example peptide analytes (peptides A through G) using transition probability matrices that had rough kinetic and conductance state characteristics resembling translocation data observed with anthrax toxin using small guest-host peptides [22] (Fig 1B). In these simulated data, there were three conductance states of the nanopore-peptide system: peptide-bound and fully blocked (state 0); peptide-bound, ~50%-conducting, and partially blocked (state 1); and fully conducting nanopore with no peptide bound (state 2). In peptides A through F, there were three species corresponding to the three conductance states (which is the simplest kinetic scheme), but in peptide G a hidden state was added, where there were two species were defined as conductance state 0 with short- and long-timescale dynamics. We wanted to include a diversity of kinetic schemes to test the generalizability of the classification algorithm. The model was designed to accurately classify peptides without prior knowledge of their kinetic schemes. These single-channel simulations were all analyzed with traditional methods by computing probabilities in state 0 and state 1 and then determining the CDF of the dwell times in state 0 and state 1 as well as the overall translocation event, fitting those distributions to exponential decays to obtain respective rate constants (Table 1). (N.B. only peptide G's state 0 dwell times deviated significantly from single-exponential fits, having a clear fast and slow phase, reflecting that it had two state 0 species in its kinetic scheme.) Approximately, 30 s of simulated data was generated for Peptides A-F, whereas 150 s was required for Peptide G to obtain similar numbers of translocation events for all peptide classes (~500 events). Class balance was further maintained by

**Table 1. Kinetic Translocation Parameters of Simulated Peptides.**

| Parameter[a] | Peptide A | Peptide B | Peptide C | Peptide D | Peptide E | Peptide F | Peptide G |
|---|---|---|---|---|---|---|---|
| $k\ (0{\to}1)\ (s^{-1})$ | 189 | 417 | 271 | 98 | 517 | 591 | 568 16[b] |
| $k\ (1{\to}0)\ (s^{-1})$ | 420 | 272 | 542 | 314 | 345 | 97 | 412 |
| $k\ (\text{Overall})\ (s^{-1})$ | 38 | 131 | 72 | 43 | 41 | 23 | 4.5 |
| $P(\text{State } 0)$ | 0.755 | 0.493 | 0.757 | 0.849 | 0.402 | 0.115 | 0.912 |
| $P(\text{State } 1)$ | 0.245 | 0.507 | 0.243 | 0.151 | 0.598 | 0.885 | 0.088 |

[a]Rate constants, $k$, and probabilities, $P$, for the indicated transitions and states, respectively, over the translocation event stream for each peptide.

[b]There are two $k$ values for the fully blocked state 0 dwells for Peptide G, corresponding to a fast and slow phase due to the presence of two different fully blocked conductance state intermediate species.

appropriate downsampling for more effective model training. Clips of the of idealized translocation event streams from these simulated datasets are shown (Fig 3).

## Peptide classification of translocation events using global features

Next, we addressed the challenging problem of accurately classifying pure test peptides from a stream of single-channel translocation events (obtained via a protein nanopore), a critical step in developing advanced biosensing technologies. To achieve this, raw translocation event streams (labeled by their conductance states from current versus time recordings) were preprocessed into state sequences representing the temporal dynamics of peptide interactions within the nanopore. In our simulated nanopore (based on anthrax toxin), there are three conductance states (which were described above); therefore, a translocation event state sequence is a series of peptide-bound state 1 or state 0 time points. Simultaneously, during the preprocessing segmentation of these translocation event state sequences, a comprehensive set of 13 event-level features, capturing local characteristics of individual translocation events, and set of global features, summarizing the overall event stream properties, were computed. These processed conductance state sequences and features were

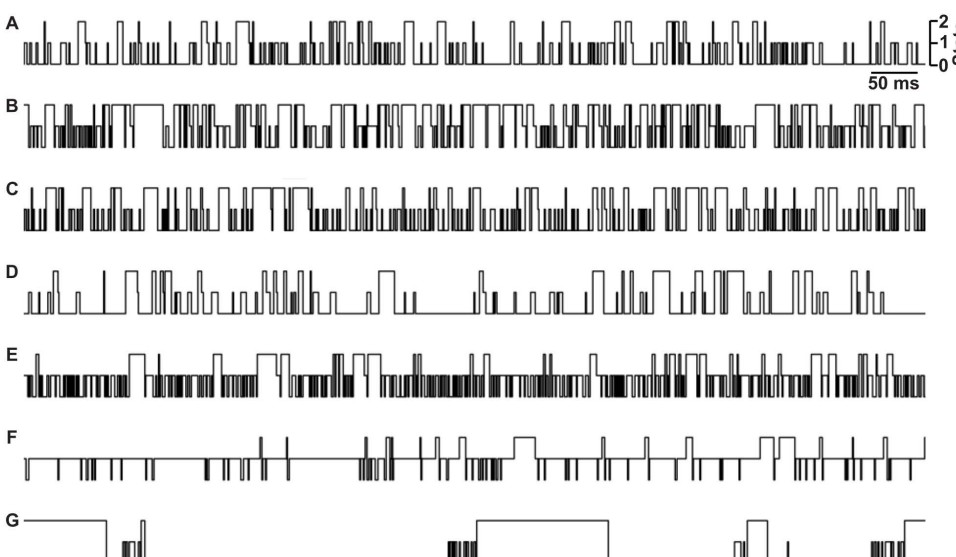

**Fig 3. Idealized nanopore conductance state versus time slices from peptide translocation simulations.** Peptide A-G are indicated at left. States are labeled as shown at top right: fully blocked (state 0), partially blocked (state 1), and fully open (state 2). Time scalebar is indicated at top right.

then fed into a hybrid NN architecture comprising a TCN [36] for state sequence processing and a dense NN for feature processing, with their outputs concatenated before final classification. (When an LSTM network was employed in place of the TCN, the model demonstrated severe validation volatility, with divergent validation behavior, as shown by both accuracy and macro-averaged F1-score, compared to the training data. This instability prevented convergence and led to poor generalization, ultimately motivating the adoption of the TCN architecture.) When using the hybrid TCN/Dense NN, the inclusion of the global feature, variance of translocation event durations, led to a significant degradation in model performance, evidenced by a drop in overall accuracy from 0.99 to 0.60 and a decrease in macro-averaged F1-score from 0.99 to 0.55, indicating that this feature introduced noise and model instability into the classification. In contrast, the model achieved exceptional performance with the remaining global features, as observed in the confusion matrix (Fig 4A), demonstrating its effectiveness in accurately classifying peptides from pure translocation event streams and highlighting the importance of careful feature selection for robust biosensor applications.

## Peptide classification training with only event-level features

We then moved to a more challenging peptide classification problem, where we tried to predict peptides based on individual translocation events (without using global features aggregated over an entire event stream). To do this, we trained the branched TCN/Dense model again using the translocation event state sequences (fed into the TCN branch) and using 12 standardized event-level features (fed into the dense branch) for Peptides A through G. (The event-level feature corresponding to the state sequence's Shannon entropy was removed from the original set of 13 features since permutation importance analysis showed it did not contribute significantly to F1-score improvement.) The model achieved an overall peptide classification accuracy of 0.70 (±0.01) on the test set (N = 6). The macro-averaged precision, recall, and F1-score were 0.70 (±0.02), 0.70 (±0.01), and 0.70 (±0.02), respectively, indicating an improved level of performance compared to previous iterations. The confusion matrix (Fig 4B) revealed a noticeable ability to classify several peptides accurately, though some degree of misclassification persisted, particularly between Peptides A and C, highlighting the challenge of

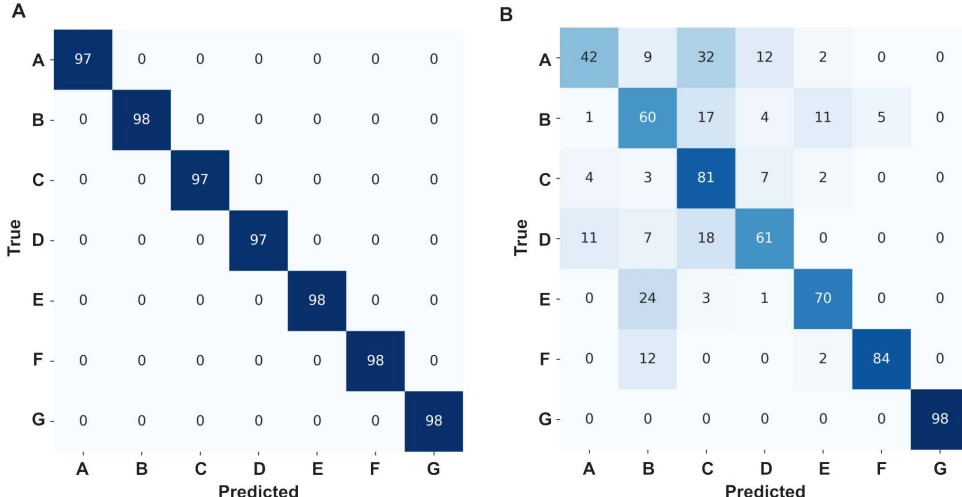

**Fig 4. Confusion matrices for TCN/Dense model peptide classifications from translocation events.** Peptides A-G are indicated when using the following inputs (A) conductance state sequences with global- and event-level features as input (overall accuracy of 0.9998 (±0.0006) and macro-averaged F1-score of 0.9998 (±0.0006)) or (B) conductance state sequences with only event-level features as input (overall accuracy of 0.70 (±0.01) and macro-averaged F1-score of 0.70 (±0.02)). The best matrix out of six replicates for either type of training is shown.

distinguishing highly similar peptides based solely on event-level kinetics due to stochasticity. Peptides D, E, F, and G generally showed higher classification performance.

**Predicting peptide mixtures from translocation events.** Leveraging the significant speed improvement (~14x faster) achieved by utilizing the GPU cores on the M4 MacBook Pro (in addition to using lower dropout rates and feature standardization), we evaluated the model's ability to predict the fractional composition of various binary mixtures and one ternary peptide mixture (Table 2). The model demonstrated varying degrees of accuracy across the different mixtures. The ternary mixture of A/C/D (40/30/30) yielded the lowest MAE of 0.045, suggesting the most accurate prediction of component fractions among the tested mixtures in terms of overall fraction estimation. Mixtures involving peptides with more distinct kinetic profiles, such as B/F (MAE: 0.084) and E/F (MAE: 0.124), showed relatively lower MAE values. However, mixtures involving peptides with more similar characteristics, such as C/D (MAE: 0.132), E/G (MAE: 0.138), B/G (MAE: 0.141), A/C (MAE: 0.143), and particularly A/D (MAE: 0.161), resulted in higher MAE values, indicating less accurate quantification of the mixture components and more pronounced misclassification between the constituent peptides. Despite the improved overall estimates of sample fractions in some mixtures, particularly A/C/D, the confusion matrices revealed persistent challenges in classifying individual events, especially between similar peptides like A, C, and D. This difficulty in accurately predicting mixture compositions at the individual translocation event level likely stems from the inherent stochasticity of single-channel kinetics and the significant overlap in the kinetic parameters exhibited by different peptides (Table 1). This overlap in features makes it challenging for the model to definitively classify individual translocation events, leading to errors in estimating the relative proportions of peptides within a mixture.

## Classification by vote totals from event-level predictions

Of course, less-reliable event-level predictions can be bolstered by gathering votes from the individual translocation event predictions over a wider stream of translocation data. Obviously, this voting-pattern tactic cannot be employed, however, when predicting mixed samples, since different peptide translocation events would be comingled in the data stream. Nonetheless, we used the voting-pattern aggregation of event-level predictions to assess performance of the TCN/Dense model on pure peptide streams. Vote totals for each peptide were collected over the entire data stream. From these voting patterns aggregated across each test peptide's translocation event stream, we assessed top1 accuracy, confidence, vote entropy, and rank for each peptide (Table 3). The vote pattern classification of pure translocation event streams demonstrated perfect Top-1 accuracy (1.0) for all individual peptides (A through G). This indicates that when predictions from all translocation events within a pure peptide stream are aggregated via a voting mechanism, the correct peptide is overwhelmingly identified as the top prediction. The average confidence across all peptides was 0.661, suggesting a reasonable level of certainty in the aggregated predictions, with some variability between peptides. Confidence was lowest for Peptide A (0.455) and highest for Peptide G (0.830). The average vote entropy was 1.426. Lower entropy values, such as those observed for Peptides F (0.936) and D (1.333), suggest a more decisive voting pattern with a larger proportion of votes concentrated on the top predicted class. Conversely, higher entropy, as seen for Peptide A (1.955) and Peptide B (1.758), indicates a more distributed voting pattern across multiple peptide classes, implying less certainty at the individual event level, even though the top vote correctly identifies the pure peptide stream. The average rank of the correct peptide was 1.0, consistent with the perfect Top-1 accuracy. These results highlight that while individual event-level predictions may be noisy, aggregating these predictions over an entire pure peptide stream effectively mitigates these uncertainties and leads to highly accurate peptide identification.

## Discussion

### Biomedical impact of peptide classification

Beyond fundamental research into peptide characteristics, the accurate and rapid classification of peptides holds significant biomedical relevance. Peptides are increasingly recognized for their therapeutic potential [37]. They serve as crucial biomarkers in a wide array of diseases, including various forms of cancer [38]. Moreover, specific peptides are emerging

**Table 2. Mixture Prediction Performance from Individual Translocation Events.**

| Peptide Mixture | Actual Fractions | Predicted Fractions | MAE | MSE | RMSE | Key Misclassifications |
|---|---|---|---|---|---|---|
| A/C | A: 0.6<br>C: 0.4 | A: 0.332<br>B: 0.08<br>C: 0.382<br>D: 0.17<br>E: 0.028<br>G: 0.008 | 0.143 | 0.036 | 0.190 | False positives (B/D/E/G)<br>Misclassifications (A→C/D, C→A/D) |
| C/D | C: 0.6<br>D: 0.4 | A: 0.166<br>B: 0.084<br>C: 0.4<br>D: 0.336<br>E: 0.014 | 0.132 | 0.022 | 0.149 | False positives (A/B/E) Misclassifications (C→A/B/D, D→A/B/C) |
| B/F | B: 0.5<br>F: 0.5 | A: 0.014<br>B: 0.41<br>C: 0.06<br>D: 0.014<br>E: 0.08<br>F: 0.422 | 0.084 | 0.007 | 0.084 | False positives (A/C/D/E)<br>Misclassifications (B→C/E/F, F→B/E) |
| E/F | E: 0.5<br>F: 0.5 | B: 0.204<br>C: 0.044<br>E: 0.354<br>F: 0.398 | 0.124 | 0.016 | 0.126 | False positives (B/C)<br>Misclassifications (E→B/C, F→B) |
| E/G | E: 0.6<br>G: 0.4 | A: 0.02<br>B: 0.164<br>C: 0.064<br>D: 0.024<br>E: 0.388<br>F: 0.004<br>G: 0.336 | 0.138 | 0.025 | 0.157 | False positives (A/B/C/D/F)<br>Misclassifications (E→B/C, G→D/A/C) |
| A/C/D | A: 0.4<br>C: 0.3<br>D: 0.3 | A: 0.3<br>B: 0.074<br>C: 0.282<br>D: 0.318<br>E: 0.018<br>G: 0.008 | 0.0453 | 0.004 | 0.060 | False Positives (B/E/G)<br>Misclassifications (A/C/D) |
| A/D | A: 0.6<br>D: 0.4 | A: 0.304<br>B: 0.068<br>C: 0.19<br>D: 0.426<br>E: 0.01<br>G: 0.002 | 0.161 | 0.044 | 0.210 | False positives (B/E/G)<br>Misclassifications (A/C/D) |
| B/G | B: 0.6<br>G: 0.4 | A: 0.026<br>B: 0.38<br>C: 0.094<br>D: 0.042<br>E: 0.084<br>F: 0.036<br>G: 0.338 | 0.141 | 0.026 | 0.162 | False positives (A/C/D/E/F) Misclassifications (B→C, G→D/C/E/A) |

as potential therapeutics, for instance, those used to treat neurodegenerative diseases [39], or as critical anti-microbials that combat infectious diseases, such as tuberculosis [40]. Furthermore, peptides play pivotal roles in immune modulation, acting as signaling molecules or therapeutic agents that can influence immune responses [41–43]. The existence of vast databases compiling key peptide biomarkers and therapeutic peptides across these fields [37,38,40,41] underscores their invaluable resource in the development of new detection and therapeutic methods. The ability to precisely identify and

**Table 3. Vote Pattern Classification of Pure Translocation Event Streams.**

| Peptide | Top1 Accuracy | Rank | Confidence | Vote Entropy |
|---|---|---|---|---|
| A | 1 | 1 | 0.455 | 1.955 |
| B | 1 | 1 | 0.597 | 1.758 |
| C | 1 | 1 | 0.602 | 1.652 |
| D | 1 | 1 | 0.728 | 1.333 |
| E | 1 | 1 | 0.635 | 1.329 |
| F | 1 | 1 | 0.779 | 0.936 |
| G | 1 | 1 | 0.830 | 1.020 |
| Mean | 1 | 1 | 0.661 | 1.426 |

differentiate individual peptides, as demonstrated by our nanopore-based classification method, is therefore a foundational step towards early disease detection, targeted therapeutic development, and personalized medicine. Our approach presents a promising platform for high-throughput peptide analysis that could contribute to advancing these critical biomedical applications.

## Nanopore peptide classifier pipeline

The overall goal here is to be able to classify peptides based on their single channel translocation events. This supervised deep learning classification problem was broken down into stages. The initial stage aimed to label for conductance state in primary multi-state peptide translocation current versus time records. A multi-layered CNN-RNN (adapted from the multi-channel classifier, Deep-Channel [28]) was used to accomplish this. In our simplified example here, there were three discrete conductance states in single-channel recordings of peptide translocations, e.g., fully blocked (state 0), partially blocked intermediate (state 1), and fully open (state 2). There is no formal limit to the number of states, but the typical states observed in peptide translocation data for the wild-type anthrax toxin PA nanopore [22] were used in the baseline model. Once a training set of many peptides' translocation data is labeled for conductance state, the second stage of the deep learning pipeline was implemented, peptide classification. To facilitate NN training, testing, and development, seven model peptides' translocation data were simulated. Our peptide classifier NN was comprised of a branched input network, where the symbolic translocation event state sequences (streams of state 0 and state 1 time points) were fed into a two-layered TCN, and a series of computed translocation event features were fed into a Dense network branch. These two branches were concatenated, and then a final output classification was determined. This implementation tested different types of classification problems, but they can be thought of as either including global features (computed over a larger event stream for a pure peptide) or only utilizing event-level features—a situation germane to classifying individual translocation events to tackle the challenge of classifying mixed peptide samples. Arguably the mixed-sample event-level prediction problem is more applicable to many peptide biosensor use cases, where targets are intermingled. While the inclusion of global features and/or the collection of votes from a wider stream of translocation events was highly successful, individual event-level prediction is more challenging, considering events are stochastic and peptides with similar biophysical parameters can be difficult to distinguish based on only a single event. Nonetheless, even with these limitations, mixed samples can be reasonably well estimated from a stream of event-level predictions.

## Event-level classification

The strategy chosen to classify peptides from single-channel recordings was to segment the data stream into individual translocation events. These events occur when the peptide was in close contact with the nanopore in a manner that interferes with the nanopore's conductance of electrolyte ions. In the example case presented here, there would be two

conductance states a peptide could be in during translocation events (state 0 or state 1). From these conductance state sequences obtained for each given translocation event, features were computed at either the most local event-level or the more global level (averaged over a wider stream of events). When event-level and global features were included along-side the state sequences, the model was highly predictive (overall accuracy of 0.9998 (±0.0006)). However, global features cannot always be obtained in all use cases, such as when mixtures of peptides are present in the sample. In these types of situations, only event-level features can be used alongside the translocation event state sequences. The resulting classification when only event-level features are used is more modest (accuracy of 0.70 (±0.01)). The degradation here is attributable to the inherent difficulty in classifying individual stochastic events, especially also when peptides share similar kinetic parameters (Table 1). This line of reasoning is supported when examining the confusion matrix for the seven peptides in event-level predictions, where similar peptides (A, C, and D) showed the greatest levels of confusion and misclassification.

### Vote aggregation over a translocation event stream

Classification of peptides using only the event-level features (as already mentioned) is more prone to error. However, when analyzing a pure peptide sample, a wider stream of event predictions can be tallied to make a more confident prediction about the identity of the peptide sample. This vote aggregation tactic produced highly accurate predictions with 100% Top-1 (Table 3). The confidence and entropy from vote totals can also be assessed to report on the quality of each prediction. In our simulated peptide data, there was higher entropy and lower confidence for A, B, and C, reflecting again the inherent noise in their individual event-level predictions. Peptides with more distinct biophysical parameters showed lower entropies and higher confidence (D, F, and G).

### The challenge of mixture predictions

Probably the most typical peptide biosensor application invariably involves mixtures of target peptides or target peptides contained in a background. To deal with these scenarios, event-level predictions using only event-level features would be required. We simulated eight different peptide mixtures to assess the classifier's performance (Table 2). Mixtures with more distinct peptides (e.g., B/F and E/F) showed lower MAE in their predictions, whereas mixtures containing A, C, and D, while able to predict the major components, were noisier with higher MAE. While the mixture predictions could identify the major components of the mixtures, a trace background of false positives emerged. We included a confidence thresh-old in our mixture predictor, which can minimize these false positives, but the best mixture predictions invariably included some false positives. Thus, it may be challenging for this predictor to identify a true minor or trace component of a peptide mixture, given the background we observe.

### Exploiting methods to enhance peptide mixture classification

While the deep learning framework presented here demonstrates high accuracy for classifying pure peptide streams and shows promising capability for peptide mixtures, the event-level classification accuracy (0.70 (±0.01)) indicates room for further improvement. Enhancing performance in this challenging area is critical for practical biosensor applications. We envision four strategies and model adjustments to address this: (i) feature engineering, (ii) exploring alternative neural network architectures, (iii) confidence-threshold optimization, and (iv) event-length filtering.

For feature engineering, future work will investigate the inclusion of additional, potentially highly discriminative, features. For instance, the exact scaled current levels of the observed conductance states, which are known to vary uniquely among different guest-host peptides in real PA nanopore data, could provide additional richer information. Furthermore, while our simulated data was simplified having only three conductance states, real guest-host peptides are known to populate four or more distinct states. Generalizing our approach to incorporate these higher numbers of states would naturally expand the feature space, potentially leading to more robust classification.

With regard to exploring alternate network architectures, initial investigation focused on TCN/Dense, chosen for its stability compared to LSTM/Dense. However, other architectures warrant exploration. CNNs, for example, may be applied directly to the scaled current traces of events and may offer powerful pattern recognition capabilities. While potentially faster to train, 1D CNNs can be more susceptible to the inherent noise in raw nanopore current data. A particularly promising avenue is the application of 1D CNNs directly to the pre-classified conductance state sequences. This approach would allow the model to learn local and contextual patterns within the sequence of discrete states, which may encode unique peptide-specific signatures that are not fully captured by aggregate features. These CNN models can also incorporate features in a concatenated Dense input layer analogous to the TCN/Dense model featured in this study.

The current event-level mixture prediction utilizes a user-defined confidence threshold. While some preliminary testing was conducted, an exhaustive, systematic optimization of this parameter across various mixture complexities and noise levels could refine the overall mixture classification accuracy. However, initial observations suggest that the primary limitation may stem from the inherent information content within very short or ambiguous individual events, rather than merely the threshold setting itself.

The current analysis did not explicitly exploit event-length filtering as a primary optimization strategy. However, we have observed in preliminary investigations that discarding translocation events below a certain duration can significantly improve classification accuracy and F1-score across various machine learning and deep learning models. This is conceptually aligned with the idea that extremely short events contain less information content, making them difficult to classify reliably (analogous to classifying an image with very few pixels). While beneficial for enhancing accuracy, this strategy must be applied judiciously, particularly for peptides known to exhibit rapid kinetics, as it could inadvertently remove a substantial portion of potentially classifiable events. A detailed study on the optimal application and trade-offs of event-length filtering will be a crucial component of our follow-up research utilizing real PA nanopore data.

## Hardware acceleration

While developing the peptide classifier, we iterated through several NN architectures, optimized hyperparameter settings, and engineered various features, where the average training time per epoch was about 6–7 minutes on an older 2014 Macbook Air. To improve performance and reduce training time, the hardware was upgraded to Apple Silicon M4 with 24 GB of RAM, and the tensorflow-metal module was installed to specifically utilize faster GPU cores. This upgrade reduced the training time per epoch to 25–30 s (~14 × improvement). As a practical matter, it is anticipated that this implementation will facilitate future training and prediction on large experimental peptide translocation datasets obtained with the anthrax toxin nanopore.

## Study limitations and future directions

This study set out to demonstrate proof of principle that nanopore biosensors can be used to classify peptides with deep learning based only on the fact that they populate a partially blocked conductance state intermediate along with the fully blocked state. Here we used a large enough controlled, simulated dataset to aid in proper model development and training, thus increasing the chances of success on adapting this approach to real-world nanopore translocation data. There were, of course, oversimplifying assumptions in our simulated single-channel peptide translocation datasets. Known peptide translocations via the PA nanopore can populate two unique partially conducting states alongside the fully blocked state during a translocation event [22] (Fig 1B); however, in our simulated peptides there was only one partially blocked state along with the fully blocked state. Also, the conductance-level of the partially blocked states can vary depending on the peptide sequence in real data [22] (Fig 1B), but in our simulated sets the partially blocked state was 50% blocked. Hence the blockade depth itself could not be exploited as a feature using our deep learning model with the simulated peptide data; however, in the future with realistic data we can certainly utilize this feature. Other limitations include the fact that the translocation event state sequences themselves were idealized and did not include any

noise beyond their inherently stochastic nature. Of course, we also learned that classifying peptides with similar kinetic parameters is also challenging, leading to cases of misclassification and confusion. We could not address this issue with global features or aggregated votes for predicting peptides mixtures, as only event-level features were applicable. The obvious future steps are to use a version of this deep learning model to predict actual peptides using the anthrax toxin nanopore. Additional partially blocked intermediates can be included in the translocation event state sequence itself, allowing for further feature engineering. Moreover, the respective blockade depths of these intermediates can vary depending on the peptide in anthrax toxin translocation data, allowing for new features to be incorporated. Nanopore engineering can also be employed to make different mutations in and around the peptide clamp constriction points (Fig 1A) to allow for multiplexed readout of target peptides; these types of mutations in the φ clamp [44] show promise in how they alter the numbers of conductance intermediates that form during peptide translocation [22]. While the TCN was used for the state sequence input (over the LSTM due to stability issues), we have not tried their combination or used other architectures, like the CNN-LSTM from Deep-Channel [28] used in our multi conductance state classifier. For the present work, downsampling was selected to provide a clear and controlled demonstration of our model's capabilities with balanced, experimentally relevant event counts, prioritizing the fidelity of the simulated data. In future work with real datasets, we may explore data augmentation of the raw current traces and apply SMOTE or ADASYN. This may potentially leverage all available data and further enhance the robustness and generalization capabilities of our classification framework, especially when dealing with naturally imbalanced real-world nanopore datasets. Mixture prediction from noisier event-level classifications (Fig 4B) may be further improved by relying on higher confidence scores or pooling similar looking classifications to average out noise.

## Conclusions

Here a nanopore peptide biosensor deep learning pipeline was described, which (i) classifies conductance state and labels raw current versus time single-channel translocation event streams and (ii) predicts peptides based on a combination of translocation conductance state sequences and their features. Predictions of peptides are bolstered and highly accurate when using global features or accumulated vote counts over a pure peptide stream. Event-level prediction used to decipher mixtures of peptides can be noisier due to the stochastic nature of single-channel data combined with the fact the peptides with similar kinetic parameters can be confused. The report here used mainly oversimplified and idealized simulated peptide translocation datasets; however, future work will be able to leverage this proof-of-concept framework to predict realistic peptides using the anthrax toxin PA nanopore as a peptide biosensor.

### Highlights

- Created nanopore biosensor peptide classification pipeline using deep learning.
- Sequences of discrete conductance state intermediates and features were learned.
- Accurate identification of pure peptide translocation streams via vote aggregation.
- Individual translocation event classifications can be used to predict peptide mixtures.

### Acknowledgments

I want to thank Richard Barrett-Jolley for stimulating discussions about Deep-Channel. I also want to thank members of the Department of Microbial Pathogenesis in the School of Dentistry at University of Maryland, Baltimore for their comments and useful discussions.

## Author contributions

**Conceptualization:** Bryan A. Krantz.

**Data curation:** Bryan A. Krantz.

**Formal analysis:** Bryan A. Krantz.

**Funding acquisition:** Bryan A. Krantz.

**Investigation:** Bryan A. Krantz.

**Methodology:** Bryan A. Krantz.

**Project administration:** Bryan A. Krantz.

**Resources:** Bryan A. Krantz.

**Software:** Bryan A. Krantz.

**Supervision:** Bryan A. Krantz.

**Validation:** Bryan A. Krantz.

**Visualization:** Bryan A. Krantz.

**Writing – original draft:** Bryan A. Krantz.

**Writing – review & editing:** Bryan A. Krantz.

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
