## [Decision Letter · Decision Letter 0]

8 Jul 2025

PONE-D-25-23818Deep Learning-Based Classification of Peptide Analytes from Single-Channel Nanopore Translocation EventsPLOS ONE

Dear Dr. Krantz,

Thank you for submitting your manuscript to PLOS ONE. After careful consideration, we feel that it has merit but does not fully meet PLOS ONE’s publication criteria as it currently stands. Therefore, we invite you to submit a revised version of the manuscript that addresses the points raised during the review process.

We look forward to receiving your revised manuscript.

Kind regards,

Salman Sadullah Usmani, Ph.D.

Academic Editor

PLOS ONE

Journal Requirements:

[National Institutes of Health (1R01AI077703 and 5R21AI124020)]. 

[This work was supported by the Department of Microbial Pathogenesis in the School of Dentistry at University of Maryland, Baltimore and by the National Institutes of Health (1R01AI077703 and 5R21AI124020).]

[National Institutes of Health (1R01AI077703 and 5R21AI124020)]

[NO].

6. We noted in your submission details that a portion of your manuscript may have been presented or published elsewhere. [The introductory Fig. 1A has an adapted stock image of the anthrax toxin nanopore. The example introductory recordings in Fig. 1B have been rescaled and reoriented from a previous publication. They are just as a demonstration and not essential to the novelty of this work or any of its conclusions.] Please clarify whether this publication was peer-reviewed and formally published. If this work was previously peer-reviewed and published, in the cover letter please provide the reason that this work does not constitute dual publication and should be included in the current manuscript.

7. When completing the data availability statement of the submission form, you indicated that you will make your data available on acceptance. We strongly recommend all authors decide on a data sharing plan before acceptance, as the process can be lengthy and hold up publication timelines. Please note that, though access restrictions are acceptable now, your entire data will need to be made freely accessible if your manuscript is accepted for publication. This policy applies to all data except where public deposition would breach compliance with the protocol approved by your research ethics board. If you are unable to adhere to our open data policy, please kindly revise your statement to explain your reasoning and we will seek the editor's input on an exemption. Please be assured that, once you have provided your new statement, the assessment of your exemption will not hold up the peer review process.

Additional Editor Comments:

Thank you for submitting your manuscript titled "Deep Learning-Based Classification of Peptide Analytes from Single-Channel Nanopore Translocation Events", which has been reviewed by two experts. Both reviewers provided constructive feedback and recognized the strength of your deep learning framework, particularly its strong performance in classifying pure peptides. I encourage you to address the key concerns raised—especially those related to feature transparency, classification of peptide mixtures, and model choice justification.

While your work is technically compelling, I also recommend adding a brief discussion on the broader biomedical relevance of peptide analyte classification. Peptides are increasingly recognized for their roles as therapeutics (PMID: 28759605), biomarkers in oncology (PMID: 28473704), and contributors to metabolic and neurodegenerative diseases (PMID: 39482312), infectious diseases (PMID: 29688365), and immune modulation (PMIDs: 30753476, 28211521, 33034338). This will help frame your work in a broader translational context.

I appreciate the innovation and effort in your study. Based on the reviewer feedback, I recommend Major Revisions before the manuscript can be considered for publication.

Reviewers' comments:

Reviewer's Responses to Questions

**Comments to the Author**

1. Is the manuscript technically sound, and do the data support the conclusions?

Reviewer #1: Yes

Reviewer #2: Yes

2. Has the statistical analysis been performed appropriately and rigorously? 

Reviewer #1: Yes

Reviewer #2: Yes

3. Have the authors made all data underlying the findings in their manuscript fully available?

Reviewer #1: Yes

Reviewer #2: Yes

4. Is the manuscript presented in an intelligible fashion and written in standard English?

Reviewer #1: Yes

Reviewer #2: Yes

5. Review Comments to the Author

Reviewer #1: The authors has commendably designed a Deep Learning-Based Classification of Peptide Analytes from Single-Channel Nanopore Translocation Events. I offer a few suggestions for potential refinement.

1. Large datasets are crucial for deep learning models, significantly enhancing their effectiveness during both training and validation phases.

2. Increasing the number of layers in a neural network can improve its predictive accuracy by enabling the model to learn more intricate and complex patterns within the data.

3. In Keras, dropout regularization can be implemented by adding a dropout layer immediately following a dense layer.

4. Training a neural network involves multiple iterations over the training data, known as epochs. Increasing the number of epochs allows the network to learn more extensively from the data, thereby potentially improving accuracy.

5. Hyperparameter optimization is an iterative process. It involves systematically experimenting with parameters such as learning rates, batch sizes, and epochs to identify the optimal combination for a specific task. Tools like grid search or random search are recommended for efficient exploration of the hyperparameter space.

Reviewer #2: This manuscript describes a deep learning-based computational pipeline developed for classifying peptide analytes based on nanopore translocation events. The study employs simulated peptide translocation data, modeling multiple discrete conductance states, processed through a two-stage neural network architecture. The initial stage classifies conductance states from raw data using a CNN-RNN network, while the subsequent stage predicts peptide identities using a Temporal Convolutional Network (TCN) and dense neural network. The approach achieves very high accuracy for classifying pure peptides (0.99) but faces challenges when classifying peptide mixtures at the event-level (0.68 accuracy). The paper demonstrates successful identification of peptides from pure samples via vote aggregation, achieving perfect accuracy, highlighting potential utility for practical nanopore-based biosensors.

Major Comments:

1. While pure peptide classification results are exceptional, mixture classification accuracy remains moderate. The authors should investigate alternative strategies or model adjustments (e.g., better feature engineering, alternative network architectures, or confidence-threshold optimization) to enhance event-level classification accuracy in mixtures.

2. It would be beneficial if the authors could provide additional details on their feature selection strategy, including how they determined which global and event-level features to include or exclude. The manuscript briefly mentions instability with the variance feature—further elaboration or supplementary analysis (e.g., ablation studies) would improve clarity and reproducibility.

3. The manuscript briefly mentions downsampling but does not explore the effects of class imbalance deeply. Given the importance of balanced classes for training robust models, it would strengthen the paper if the authors discussed alternative methods such as data augmentation or synthetic data generation for class balancing.

4. The choice of TCN over LSTM due to stability issues is mentioned but not fully justified with quantitative comparison data. Including comparisons or figures illustrating the performance/stability differences between TCN and LSTM networks would be valuable.

Minor Commnets:

1. Fig. 2 (Page 33): Clarify the legend by explicitly labeling the color codes or state numbers to improve readability.

2. Fig. 4 (Page 35): Consider providing numerical accuracy or F1-scores within the confusion matrices themselves for greater clarity.

3. On page 7, the authors state that source code, simulated datasets, and real datasets are available on GitHub but the link is not included. Please ensure the link is clearly provided for reviewers and readers.

4. The manuscript reports accuracy, precision, recall, and F1-score but does not consistently include confidence intervals or statistical significance. Including such metrics would improve the scientific rigor and clarity of results.

5. Please check the consistency of formatting (e.g., space between "Deep-Channel" and subsequent terms).

6. The manuscript clearly highlights the performance gain from hardware upgrade. However, it would be helpful to briefly discuss the scalability or potential limitations of this pipeline in practical scenarios involving significantly larger datasets.

7. Please read the manuscript carefully, and rectify the grammatical and punctuation issues.

6. PLOS authors have the option to publish the peer review history of their article (what does this mean? ). If published, this will include your full peer review and any attached files.

**Do you want your identity to be public for this peer review?** For information about this choice, including consent withdrawal, please see our Privacy Policy .

Reviewer #1: No

Reviewer #2: **Yes: ** SUMEET PATIYAL

---

## [Author Response · Author response to Decision Letter 1]

22 Jul 2025

Editor's Comments/Journal Requirements:

1. [Editor/Journal Requirement 1] Ensure the manuscript meets PLOS ONE's style requirements, including file naming. (Refer to provided templates).

I added Headings Styles in all the proper places and followed other stylistic conventions to the manuscript by the Style Guidelines.

2. [Editor/Journal Requirement 2] Ensure all author-generated code is made available without restrictions upon publication, following PLOS ONE's code sharing guidelines. Provide the GitHub link for reviewers/readers.

The author-generated code was added to GiHub at the url (https://github.com/bakrantz/Pept-Class). Also note that the experimental and simulated datasets (labeled stream .csv and processed events .pkl files) are contained there alongside the Python code.

3. [Editor/Journal Requirement 3] Amend the financial disclosure statement to explicitly state the funders' role in the study (e.g., "The funders had no role in study design, data collection and analysis, decision to publish, or preparation of the manuscript." or an accurate alternative). Include this amended statement in the cover letter.

I included the funding statement in the cover letter.

4. [Editor/Journal Requirement 4] Remove all funding-related text from the Acknowledgments section or other areas of the manuscript. Confirm the preferred updated Funding Statement in the cover letter.

I removed funding-related text from the Acknowledgments.

5. [Editor/Journal Requirement 5] Complete the Competing Interests on the online submission form. If no competing interests, state "The authors have declared that no competing interests exist." in the cover letter.

I added the statement: “The authors have declared that no competing interests exist” to the cover letter.

6. [Editor/Journal Requirement 6] Clarify whether the adapted/rescaled/reoriented figures (Fig. 1A and 1B) were from previously peer-reviewed and formally published work. If so, provide the reason in the cover letter why this does not constitute dual publication.

Fig 1 is a background/introduction figure, which is purely pedagogical, and does not constitute the main examined Results data and findings of this manuscript. Instead of re-using the bitmap in that figure, I re-generated a new bitmap in Fig 1A with slightly different orientation and colors. Therefore, the purpose of the image is intact but there are no re-use/licensing concerns for PLOS. In Fig 1B, instead of re-plotting earlier published data, I decided to re-make the figure with new peptide translocation events samples, thus preserving the pedagogical purpose of showing that a nanopore system indeed populates several intermediate conductance states. Hopefully this satisfies any lingering concerns.

7. [Editor/Journal Requirement 7] Decide on a data sharing plan now, rather than waiting for acceptance. Ensure that data will be made freely accessible upon publication, explaining any necessary access restrictions or seeking an exemption if public deposition is not feasible.

As I stated above regarding the generated code deposition at GitHub, all datasets (both simulated and experimental) are deposited at GitHub (https://github.com/bakrantz/Pept-Class). This will be more convenient to have data and code together.

8. [Editor/Editor Comment] Add a brief discussion on the broader biomedical relevance of peptide analyte classification, including its roles as therapeutics, biomarkers, and contributors to various diseases (metabolic, neurodegenerative, infectious, immune modulation).

I created a new subsection at the start of the Discussion to frame the broader impacts.

Biomedical impact of peptide classification

Beyond fundamental research into peptide characteristics, the accurate and rapid classification of peptides holds significant biomedical relevance. Peptides are increasingly recognized for their therapeutic potential [1]. They serve as crucial biomarkers in a wide array of diseases, including various forms of cancer [2]. Moreover, specific peptides are emerging as potential therapeutics, for instance, those used to treat neurodegenerative diseases [3], or as critical anti-microbials that combat infectious diseases, such as tuberculosis [4]. Furthermore, peptides play pivotal roles in immune modulation, acting as signaling molecules or therapeutic agents that can influence immune responses [5-7]. The existence of vast databases compiling key peptide biomarkers and therapeutic peptides across these fields [1, 2, 4, 5] underscores their invaluable resource in the development of new detection and therapeutic methods. The ability to precisely identify and differentiate individual peptides, as demonstrated by our nanopore-based classification method, is therefore a foundational step towards early disease detection, targeted therapeutic development, and personalized medicine. Our approach presents a promising platform for high-throughput peptide analysis that could contribute to advancing these critical biomedical applications.

Note the citations are of course different numbers in the text of the manuscript. They are enumerated here for your reference:

1. Usmani SS, Bedi G, Samuel JS, Singh S, Kalra S, Kumar P, et al. THPdb: Database of FDA-approved peptide and protein therapeutics. PLoS One. 2017;12(7):e0181748. Epub 20170731. doi: 10.1371/journal.pone.0181748. PubMed PMID: 28759605; PubMed Central PMCID: PMCPMC5536290.

2. Bhalla S, Verma R, Kaur H, Kumar R, Usmani SS, Sharma S, et al. CancerPDF: A repository of cancer-associated peptidome found in human biofluids. Sci Rep. 2017;7(1):1511. Epub 20170504. doi: 10.1038/s41598-017-01633-3. PubMed PMID: 28473704; PubMed Central PMCID: PMCPMC5431423.

3. Usmani SS, Jung HG, Zhang Q, Kim MW, Choi Y, Caglayan AB, et al. Targeting the hypothalamus for modeling age-related DNA methylation and developing OXT-GnRH combinational therapy against Alzheimer's disease-like pathologies in male mouse model. Nat Commun. 2024;15(1):9419. Epub 20241031. doi: 10.1038/s41467-024-53507-8. PubMed PMID: 39482312; PubMed Central PMCID: PMCPMC11528003.

4. Usmani SS, Kumar R, Kumar V, Singh S, Raghava GPS. AntiTbPdb: a knowledgebase of anti-tubercular peptides. Database (Oxford). 2018;2018. doi: 10.1093/database/bay025. PubMed PMID: 29688365; PubMed Central PMCID: PMCPMC5829563.

5. Usmani SS, Agrawal P, Sehgal M, Patel PK, Raghava GPS. ImmunoSPdb: an archive of immunosuppressive peptides. Database (Oxford). 2019;2019. Epub 20190101. doi: 10.1093/database/baz012. PubMed PMID: 30753476; PubMed Central PMCID: PMCPMC6367516.

6. Nagpal G, Usmani SS, Dhanda SK, Kaur H, Singh S, Sharma M, et al. Computer-aided designing of immunosuppressive peptides based on IL-10 inducing potential. Sci Rep. 2017;7:42851. Epub 20170217. doi: 10.1038/srep42851. PubMed PMID: 28211521; PubMed Central PMCID: PMCPMC5314457.

7. Dhall A, Patiyal S, Sharma N, Usmani SS, Raghava GPS. Computer-aided prediction and design of IL-6 inducing peptides: IL-6 plays a crucial role in COVID-19. Brief Bioinform. 2021;22(2):936–45. doi: 10.1093/bib/bbaa259. PubMed PMID: 33034338; PubMed Central PMCID: PMCPMC7665369.

Reviewer #1 Comments:

1. [Reviewer #1, Item 1] Discuss or elaborate on how large datasets are crucial for deep learning models and how they enhance effectiveness during training and validation.

Thank you for highlighting this fundamental principle, which we alluded to in the introduction. I fully agree, and I decided to at first use simulated data to generate a large and controlled dataset rather than use a smaller existing sample size of experimental data; increased data volume is critical for training robust deep learning models. We added some phrasing to the manuscript in the Methods and Discussion to further emphasize this concern in our study design.

In the Methods: “Simulated data was chosen in this study to have a large enough controlled dataset to train robust deep learning models and facilitate their development.”

In the Discussion: “Here we used a large enough controlled, simulated dataset to aid in proper model development and training, thus increasing the chances of success on adapting this approach to real-world nanopore translocation data.”

2. [Reviewer #1, Item 2] Discuss or elaborate on how increasing the number of layers in a neural network can improve predictive accuracy by learning more intricate patterns.

The model architecture (TCN-Dense) settled on was designed with a specific number of layers chosen after preliminary architectural exploration to balance complexity, training stability, and performance. The training time was also optimized by reducing dilations and filters without sacrificing evaluation metrics and performance. There are trade-offs in increasing model complexity (e.g., longer training times and overfitting risks).

We added this sentence to the methods explaining how we settled on the number of TCN blocks and layers as well as other key parameters: “The number of TCN blocks, dilations, filters and kernel size were chosen to balance the ability to maintain evaluation metrics while reducing computational costs.”

3. [Reviewer #1, Item 3] Ensure proper implementation and discussion of dropout regularization in the Keras model.

The dropout_rate parameter was applied in the TCN blocks at a level of 0.3. Dropout regularization was also applied in the Dense layers at a level of 0.3. These dropouts were specified in the Methods section. We added a statement to the Methods reflecting that dropout regularization reduces overfitting.

4. [Reviewer #1, Item 4] Discuss or elaborate on the role of increasing epochs in training and how it can improve accuracy.

We trained the models for a sufficient number of epochs, employing callbacks of early stopping with a patience to prevent overfitting and ensure optimal learning without unnecessary computation. The maximum number of epochs allowed was 30 for TCN-Dense model. The validation metrics were largely plateaued at that point. No further optimization of epoch number was performed.

The manuscript describes our implementation of this in the Methods: “Training was conducted with a batch size of 32 for 30 epochs, with 20% of the training data reserved for validation. To prevent overfitting and optimize training, early stopping (patience of 20 epochs), model checkpointing (saving the best weights based on validation loss), and a learning rate reduction on plateau (factor of 0.5, patience of 5 epochs, minimum learning rate of 10-5) were implemented as callbacks during the training process.”

5. [Reviewer #1, Item 5] Discuss hyperparameter optimization as an iterative process, mentioning systematic experimentation with learning rates, batch sizes, and epochs, and recommending tools like grid search or random search for efficient exploration.

We acknowledge the reviewer's emphasis on systematic hyperparameter optimization. While we did not employ an automated grid search or extensive random search for all hyperparameters, our model development involved an iterative and principled manual tuning process. We systematically explored the impact of key architectural choices, including TCN block arrangements, filter numbers, dilation rates, and kernel sizes, on model performance and stability. Similarly, we fine-tuned the dropout rate by observing its effect on overfitting and learning efficiency. We basically moved dropout until it went from the extreme of overfitting to the other extreme of not learning effectively, striving for good balance. This iterative approach allowed us to identify the optimal parameter combinations that yielded the robust performance reported in our study. We agree that formal search methodologies could be beneficial for even broader parameter space exploration in future investigations.

We added the following clarification of our more manual parameter search process to the Methods section: “The selection of the TCN/Dense architecture's specific parameters, including the number and configuration of TCN blocks, filter counts, dilation rates, and kernel sizes, as well as the dropout rate, was refined through an iterative manual tuning process. This involved systematic experimentation and evaluation on the validation dataset to optimize for classification performance and training stability.”

Reviewer #2 Comments:

Major Comments:

1. [Reviewer #2, Major 1] Investigate and discuss alternative strategies or model adjustments to enhance event-level classification accuracy in peptide mixtures (e.g., better feature engineering, alternative network architectures, or confidence-threshold optimization).

We appreciate the reviewer's insightful comment regarding the moderate event-level classification accuracy (0.70) for peptide mixtures. We acknowledge that this is a key area for improvement and have thoroughly investigated potential strategies and model adjustments. We've added a dedicated section in the Discussion to detail these avenues, outlining their mechanisms and our preliminary thoughts or prior findings, and framing deeper explorations as important future work, particularly with real nanopore datasets.

Here is a breakdown of the areas suggested by the reviewer to explore as well as another area we have been investigating more recently:

A. Enhanced Feature Engineering. We agree that expanding our feature set holds significant promise. As discussed in the manuscript, the inclusion of scaled current levels of observed conductance states, which are known to vary between guest-host peptides in real PA nanopore data, could provide richer information. Our simulated three-state data, while demonstrating feasibility, represents a simplified scenario. Generalizing to peptides that populate four or more conductance states in real systems would naturally introduce more discriminative features that could enhance mixture classification.

B. Alternative Network Architectures. While our current manuscript details the stability advantages of TCN-Dense over LSTM-Dense, we recognize that other architectures could be beneficial. We have considered and, in some preliminary testing since initial submission, explored CNN-Dense architectures for scaled current traces of the translocation events. While powerful and often faster to train, they can be more susceptible to noise. A particularly promising future direction is the application of 1D CNNs directly to the state sequences themselves. This would allow the model to learn local patterns within the state transitions, which could be highly informative.

C. Confidence-Threshold Optimization. The confidence threshold is a user-defined parameter in our event-level mixture prediction script. We did perform some initial exploration of this parameter and found that while it modestly affected predictions, extremely selective thresholds did not universally lead to more faithful predictors across all mixtures. This suggests that the primary limitation may lie in the inherent information content of individual short events or the features themselves, rather than merely the threshold. A more in-depth, systematic optimization would be warranted in future studies.

D. Event-Length Filtering. Although not explicitly detailed in the initial submission, our subsequent preliminary investigations have shown that filtering out very short-duration events can indeed improve classification accuracy and F1-score across various neural network and traditional machine learning models. Conceptually, short events contain less information. While effective, this approach must be applied judiciously, especially for peptides with fast kinetics where many informative events could be inadvertently removed. We deem this a critical strategy to explore in greater depth as part of our follow-up study on real nanopore event datasets, where event duration distributions are empirically known.

These strategies collectively offer compelling avenues for enhancing event-level classification accuracy in complex peptide mixtures, forming a significant part of our future research.

Because this is an important topic. We included a section in the Discussion, delineating these research prospects in improving event-level predictions of mixed samples.

“Exploiting methods to enhance peptide mixture classification

While the deep learning framework presented here demonstrates high

---

## [Decision Letter · Decision Letter 1]

26 Aug 2025

Deep learning-based classification of peptide analytes from single-channel nanopore translocation events

PONE-D-25-23818R1

Dear Dr. Krantz,

We’re pleased to inform you that your manuscript has been judged scientifically suitable for publication and will be formally accepted for publication once it meets all outstanding technical requirements.

Kind regards,

Salman Sadullah Usmani, Ph.D.

Academic Editor

PLOS ONE

Additional Editor Comments (optional):

Reviewers' comments:

Reviewer's Responses to Questions

**Comments to the Author**

1. If the authors have adequately addressed your comments raised in a previous round of review and you feel that this manuscript is now acceptable for publication, you may indicate that here to bypass the “Comments to the Author” section, enter your conflict of interest statement in the “Confidential to Editor” section, and submit your "Accept" recommendation.

Reviewer #1: All comments have been addressed

Reviewer #2: All comments have been addressed

2. Is the manuscript technically sound, and do the data support the conclusions?

Reviewer #1: Yes

Reviewer #2: Yes

3. Has the statistical analysis been performed appropriately and rigorously? 

Reviewer #1: (No Response)

Reviewer #2: Yes

4. Have the authors made all data underlying the findings in their manuscript fully available?

Reviewer #1: Yes

Reviewer #2: Yes

5. Is the manuscript presented in an intelligible fashion and written in standard English?

Reviewer #1: Yes

Reviewer #2: Yes

6. Review Comments to the Author

Reviewer #1: Author has address all the review comments that I have added in first review. I would like to propose the acceptance of this paper.

Reviewer #2: The authors have addressed all my comments satisfactorily. hence, I have no further comments. Kudos for the great work.

7. PLOS authors have the option to publish the peer review history of their article (what does this mean? ). If published, this will include your full peer review and any attached files.

**Do you want your identity to be public for this peer review?** For information about this choice, including consent withdrawal, please see our Privacy Policy .

Reviewer #1: **Yes: ** Vikram Yadav

Reviewer #2: **Yes: ** SUMEET PATIYAL

---

## [Editor Report · Acceptance letter]

PONE-D-25-23818R1

PLOS ONE

Dear Dr. Krantz,

I'm pleased to inform you that your manuscript has been deemed suitable for publication in PLOS ONE. Congratulations! Your manuscript is now being handed over to our production team.

Kind regards,

on behalf of

Dr. Salman Sadullah Usmani

Academic Editor

PLOS ONE